# Comparison Study of the Two Biometers Based on Swept-Source Optical Coherence Tomography Technology

**DOI:** 10.3390/diagnostics12030598

**Published:** 2022-02-26

**Authors:** Jing Dong, Jinhan Yao, Shuimiao Chang, Piotr Kanclerz, Ramin Khoramnia, Xiaogang Wang

**Affiliations:** 1Department of Ophthalmology, First Hospital of Shanxi Medical University, Taiyuan 030001, China; dongjing790610@163.com; 2Department of Cataract, Shanxi Eye Hospital Affiliated to Shanxi Medical University, Taiyuan 030002, China; yjh961106@163.com (J.Y.); changsm991101@163.com (S.C.); 3Hygeia Clinic, 80-286 Gdańsk, Poland; p.kanclerz@gumed.edu.pl; 4Helsinki Retina Research Group, University of Helsinki, 00014 Helsinki, Finland; 5The David J. Apple International Laboratory for Ocular Pathology, Department of Ophthalmology, University of Heidelberg, 69120 Heidelberg, Germany; ramin.khoramnia@med.uni-heidelberg.de

**Keywords:** swept-source optical coherence tomography, axial length, astigmatism, acquired rate, anterior chamber depth

## Abstract

This research aimed to investigate the potential differences in the parameters, including axial length (AL), central corneal thickness (CCT), anterior chamber depth (ACD), lens thickness (LT), flat keratometry (Kf), steep keratometry (Ks), mean keratometry (Km), astigmatism, white-to-white (WTW) distance, acquired rate, and intraocular lens (IOL) power, between the two swept-source optical coherence tomography (SS-OCT) biometers, the ANTERION (biometer A) and IOLMaster 700 (biometer B). In a prospective observational comparative case series study, we enrolled 198 eyes undergoing cataract surgery. The AL, CCT, ACD, LT, Kf, Ks, Km, astigmatism, WTW, acquired rate, and IOL power were assessed. McNemar tests compared the acquired rate, and the paired sample *t*-test compared the quantitative measurement results between the groups. Nineteen eyes were excluded owing to missing AL data for either biometer. Finally, data from 179 eyes were analyzed. Between the two devices, no significant difference was found in AL, astigmatism magnitude, J0, and J45, while significant differences existed in CCT, ACD, LT, Kf, Ks, Km, WTW, astigmatism axis, and IOL power; no statistical significance was found in the AL acquired rate (biometer A, 90.9% and biometer B, 93.9%). Approximately 65.4% of eyes demonstrated ≥0.5-D difference in IOL power between the two biometers. In conclusion, the two biometers showed significant differences in all measurements (CCT, ACD, LT, K, WTW, astigmatism axis, and IOL power), except for AL.

## 1. Introduction

With the development of refractive cataract surgery and premium intraocular lenses (IOL), precise and accurate biometric measurements play an important role in IOL calculations. Compared with ultrasonographic A-scan and partial coherence interferometry technology, the current swept-source optical coherence tomography (SS-OCT) technique provides a higher accuracy and acquired rate in biometric measurement in clinics [1,2,3,4,5].

The ANTERION SS-OCT biometer (i.e., biometer A; Heidelberg Engineering GmbH, Heidelberg, Germany) utilizes a 1300-nm wavelength light source, captures a bigger scanning width (16.5 mm) and deeper scanning depth (14.5 mm), and its measuring range for the axial length (AL) is 14–32 mm [5].

Different from ANTERION, IOLMaster 700 (i.e., biometer B; Carl Zeiss Meditec AG, Jena, Germany) SS-OCT biometry uses a 1060-nm wavelength as the light source, which is a shorter wavelength than that of ANTERION. Moreover, it can measure the biometric data and provide a cross-sectional image of a small central macular area to testify the patient’s fixation during data capture [6].

A recent study from Panthier et al. [7] demonstrated that the two biometers (ANTERION and IOLMaster 700) provided good repeatability and high correlation for the anterior parameter measurements, such as AL, mean keratometry (Km), central corneal thickness (CCT), anterior chamber depth (ACD), lens thickness (LT), and white-to-white (WTW) distance, and all these parameters were not interchangeable. However, astigmatism and the calculated IOL power were not compared in their study. As a supplement to the abovementioned study, this study aimed to determine the correlation, consistency, and level of agreement of other ocular parameters between the two biometers.

## 2. Materials and Methods

### 2.1. Participants

This prospective cross-sectional study enrolled consecutive patients who underwent cataract surgery at the Shanxi Eye Hospital, affiliated with Shanxi Medical University (Taiyuan, Shanxi, China), between November 2020 and October 2021. The inclusion criteria were as follows: no systemic disease; no pathological alteration of the anterior segment (such as keratoconus, zonular dialysis, pseudoexfoliation syndrome, and corneal opacity); no retinal diseases impairing visual function; and no previous anterior or posterior segment surgery. Patients who could not cooperate with the data capturing procedure and failed to pass the image quality check were excluded.

The research protocol was approved by the Institutional Review Board of Shanxi Medical University (No. 2019LL130) and conducted according to the tenets of the Declaration of Helsinki. Written informed consent was obtained from each participant after explaining the nature of this study. This observational study has been registered online (International Standard Randomized Controlled Trials. Available online: http://www.controlled-trials.com (accessed on 8 November 2021) with the registration number: ISRCTN13860301.

### 2.2. Sample Size

The sample size for the paired samples *t*-test was calculated using MedCalc software (Version 20.014, MedCalc Softwase Ltd., Ostend, Belgium). The type I error (Alpha, Significance) was set as 0.05, and the type II error (Beta, 1-Power) was set as 0.20. Based on previous AL comparison results from Shetty et al. [8], the input value of the mean AL difference was 0.01, and the standard deviation of AL differences was 0.03. After the calculations, the minimum required number was 73 eyes.

### 2.3. Data Acquisition

All patients received biometric data captured with the sequence of biometer A and then with that of biometer B in mesopic conditions without pupil dilation (Figure 1). All images were captured by the same experienced ophthalmologist for each biometer (biometer A: XGW and biometer B: ZJJ). The software versions for biometer A and B were 1.3.4.0 and 1.88.1.64861, respectively.

### 2.4. Astigmatism Vector Analysis and Double-Angle Plots of Astigmatism

The astigmatism magnitude and axis measured by the two devices were compared using the power vector analysis method by Thibos et al. [9]. The astigmatism values were converted into rectangular vectors J0 and J45 for the final data comparison. Compared to the single-angle plots, the double-angle plots can not only appropriately demonstrate the magnitude and axis of the average astigmatism (the centroid) and the confidence ellipse but also maintain the spatial relationship of every astigmatism value. Therefore, the double-angle plots were plotted using the tools and methods created by Abulafia et al. [10].

### 2.5. Intraocular Lens Power Calculation

The IOL power was calculated using the online Barrett Universal II Formula calculator with all the parameter (including optional parameters) inputs and a K index of 1.3375 (Barrett Universal II Formula calculator. Available online: https://calc.apacrs.org/barrett_universal2105/ (accessed on 10 October 2021). Bausch & Lomb MX60 was selected as the IOL type with a lens factor of 1.99 and an A constant of 119.2 for the calculations. Target refraction was set for Plano.

### 2.6. Statistical Analyses

Statistical analyses were performed using a commercial software (Statistical Package for the Social Sciences (SPSS) version 13.0; Chicago, IL, USA, SPSS Inc.). Normality distribution was testified using the Shapiro–Wilk test, and the paired sample *t*-test was performed to compare the quantitative measurements in the normally distributed data. McNemar’s test was performed to compare the acquisition rates between groups. Further, Pearson’s correlation coefficient was performed to estimate the correlation between the two biometers. The limit of agreement (LoA) of the Bland–Altman method was used to assess the agreement and potential systematic differences between the two biometers. All tests had a significance level of 5%.

## 3. Results

In total, 198 eyes of 145 patients (67 male, 78 female) were included in the study. The mean age was 67 ± 14 (range, 19–92) years. For AL measurements, biometer A failed in 18 eyes, and biometer B failed in 12 eyes (including 11 eyes that failed in both biometers). Cases of failed capture were mainly mature cataract with a Lens Opacity Classification System III cortical score; the nuclear grades were C5N4/5 or more. Therefore, the capture rates of biometers A and B were 90.9% (180/198) and 93.9% (186/198), respectively. Moreover, the technical failure rate was not different between the two devices (*p* = 0.070).

Finally, 179 eyes with complete biometric data for both devices were included in the final data comparison. On comparing astigmatism, no significant difference was found in the astigmatism magnitude, J0, and J45 between the two devices (Figure 2, all *p* > 0.439), but a significant difference was found for the astigmatism axis comparison (Table 1, *p* = 0.009). For 142 eyes with astigmatism >0.5 diopter (D), as measured using biometer A, biometer A demonstrated about 9 degrees lower values for the astigmatism axis in comparison with biometer B (*p* = 0.026).

No statistical difference was found in the AL measurements for the two biometers (Table 2, *p* = 0.469). Significant differences were observed in CCT, ACD, LT, Kf, Ks, Km, and WTW distance between the two devices (Table 2, all *p* < 0.001). Significant correlations were found for all of the above-mentioned nine parameters (Figure 3 and Table 3, all *p* < 0.001). The mean difference and the 95% limits of agreement were shown in the Bland–Altman plots for the nine parameters (Figure 4).

The mean IOL power difference was about 0.34 D between the two devices with the online Barrett calculator (Table 2, *p* < 0.001). In total, 62 (34.6%) eyes showed no difference in IOL power calculation. However, approximately 65.4% (117 eyes) of the eyes demonstrated ≥ 0.5 D difference in IOL power between the two biometers.

## 4. Discussion

In our study, the major findings were as follows: (1) Both two biometers provided a comparable AL acquired rate for patients with cataract, and (2) based on the vector analysis, no significant difference was found between the anterior corneal astigmatism measurement between the two biometers. However, the astigmatism axis values were different. (3) AL measurement data were interchangeable between the two biometers; (4) the other anterior parameters, such as CCT, Ks, Kf, Km, ACD, LT, and WTW distance, were not interchangeable; and (5) a high percentage (65.4%) of IOL power difference > 0.5 diopters existed between the two biometers.

Different from Panthier et al.’s [7] study, where the IOLMaster 700 demonstrated a significantly higher AL measurement rate than ANTERION (100% and 95.2%, respectively), the AL measurement failure rate was not significantly different between the two devices in the current study, but biometer A failed to measure AL in six more eyes than biometer B. This potential discrepancy may be because of the different AL acquisition methods, with averaging three consecutive subsets of data for biometer A and averaging the values of three scans in each of the six meridians for biometer B [7].

No significant difference was found in the AL measurements between the two biometers, and the average difference was 0.002 mm; this was in concordance with previous studies (mean difference ranges from −0.0044 to −0.04) (Table 4) [7,8,11,12,13,14,15]. A 0.002-mm AL difference may result in an approximately 0.005-D difference of IOL power, which is not clinically relevant. Therefore, AL data were interchangeable for these two biometers in this study. 

Similar to previous studies, we also found significant differences in the Ks, Kf, and Km measurements between the two devices [7,8,14]. Moreover, all keratometry values from biometer B were ~0.2 D higher than those from biometer A. Specifically, when comparing Km, both the right and left eyes showed the same tendency, and it was about 0.23 D flatter using biometer A than biometer B; these results are in contrast to those of Panthier et al. [7], who showed that biometer A measured the Km to be 0.11 D flatter in both the eyes, 0.2 D flatter in right eyes, and 0.07 D steeper in left eyes when compared with biometer B. These noticeable discrepancies were most likely related to the variations in sample size, measurement technique, and measurement diameter. Biometer B calculated the anterior corneal keratometry data from 18 reference points in the keratometry image using hexagonal patterns at approximately the 1.5-, 2.5-, and 3.5-mm optical zones, and the keratometry reading of the 2.5-mm optical zone was used for the IOL power calculations and for the comparisons in this study [16]. However, biometer A measured the anterior corneal keratometry values based on corneal topography in a 3.0-mm optical zone with a 65-radial scan pattern [5].

No significant difference was found between the two biometers for the anterior corneal astigmatism comparison using the vector analysis or only for the comparison of the magnitude of the astigmatism, but the astigmatism axis demonstrated approximately 9° difference for the eyes with astigmatism > 0.5 D. Based on the common statement relating the misalignment or rotation of toric IOL to lose 3.3% of its astigmatism correction effect per degree, 9° could cause a significant astigmatism correction difference in clinical practice [17].

Similar to previous studies, a significant difference was found in the ACD measurements. Biometer A provided ~0.07-mm higher mean values than biometer B [7,8,11,12,13,14,15]. Moreover, a 95% LoA was found between 0.17 mm and −0.03 mm with good agreement. This potential difference was minimal but may affect the postoperative ACD estimations and further affect the IOL calculation results [18].

The mean difference in LT was 0.04 mm, which was in the range of those published values of previous studies that ranged from −0.06 to 0.154 mm [7,8,11,12,13,14,15]. This potential difference between the two biometers may affect the postoperative IOL position, especially for some IOL formulas (Olsen, Holladay 2, and Barrett formulas) using it as required or optional input of the parameters [18,19,20].

Similar to other studies, biometer A provided ~8-μm thinner CCT values than biometer B in the present study [7,8,12,13]. This difference may be clinically relevant in glaucoma screening, preoperative evaluations for corneal refractive surgery, or IOL formulas using CCT as a variable (Olsen and Kane formulas) [21,22,23].

All the above-mentioned biometric measurement discrepancies between the two devices may also be caused by differences in the operating wavelengths (1300-nm wavelength imaging demonstrating strong absorption by aqueous humor and vitreous humor and 1060 nm imaging device demonstrating the least absorption relative to 1300 nm and nearly zero water dispersion) and the difference in the wavenumber calibration and dispersion compensation schemes adopted in each device [24,25]. 

Due to the popularity of phakic IOL implantation, WTW distance remains a vital biometric parameter for its diameter calculations [26]. Moreover, some IOL formulas (Barrett and Holladay 2 formulas) also consider this parameter a variable [20]. For WTW distance measurements, the two biometers demonstrated a ~0.24 mm (95% LoA, −0.83 to 0.35 mm) clinically significant difference, and the IOLMaster 700 always provided greater values, similar to previous studies [7,8,12,15]. The potential disparity between the two biometers should be related to the fundamental image acquiring methods and analysis methods.

To compare the combined effects of the above data for IOL power calculations, we used the widely used online Barrett calculator to find the potential differences between the two biometers, with unexpected results. Both our study and the lone previous study by Shetty et al. [8] demonstrated significant differences for the mean IOL power calculation comparison using this calculator. However, the mean difference we obtained was greater than in their study (0.34 D and −0.19 D). For IOL intervals of 0.25 D (e.g., softec HD IOL), this finding may be clinically meaningful. Moreover, approximately 65.4% (117 eyes) of the eyes demonstrated IOL power differences higher than 0.5 D (including 27 eyes with 1.0-D difference and three eyes with 1.5-D difference). When we used the formula IOL prediction error = 0.7 × IOL power error, we found that 67.2% (127 eyes) still showed ≥|0.35| D difference, including 15.9% (30 eyes), with ≥|0.70| D difference [27]. This finding emphasizes that the composite effect of all the input parameters should be taken into consideration for IOL power calculations, not just a single factor to make the conclusion.

### Strengths and Limitations

As a supplement to previous studies, this study determines the correlation, consistency, and level of agreement of more ocular parameters between the two biometers. Consequently, these results will allow us to determine the ocular parameter interchangeability between the IOLMaster 700 and ANTERION. 

Our study has some limitations. First, both the eyes of 53 patients were enrolled in this comparative study, which may neglect the potential existing correlations between the two eyes of the same participant [28]. Based on Panthier et al.’s [7] study, comparisons of a single eye or both eyes did not change the results. Therefore, we believe that our results are also reliable. Second, because of only 14 short eyes (AL < 22.0 mm) and 18 long eyes (AL > 26.0 mm) in the whole dataset, there were no subgroups of short, normal, and long AL to test the differences in each subgroup. Third, we did not compare the total keratometry and total astigmatism, because the current version of IOLMaster 700 in our clinic did not provide these data. Fourth, the postoperative refractive outcomes data to test the IOL calculation accuracy between the two biometers were not available in the current study. Therefore, a more detailed comparison should be considered in future studies. However, we still believe that the current study results could provide useful information for clinical references.

## 5. Conclusions

In conclusion, the potential differences in CCT, ACD, LT, K, WTW, astigmatism axis, and, especially, a high rate of IOL power difference between the two SS-OCT biometers should be considered in clinical practice.

## Figures and Tables

**Figure 1 diagnostics-12-00598-f001:**
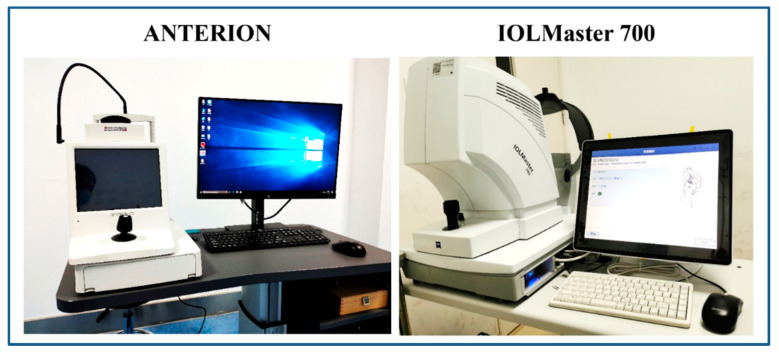
The two devices used in this study.

**Figure 2 diagnostics-12-00598-f002:**
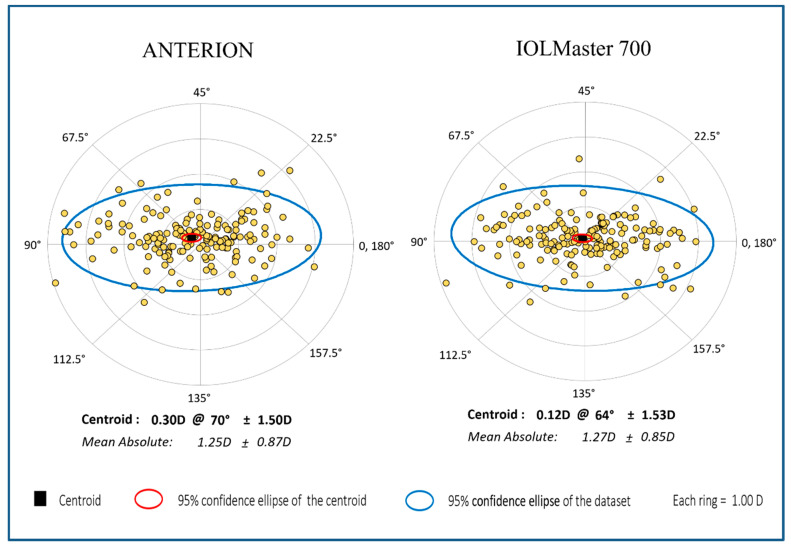
The double-angle plot of anterior corneal astigmatism of ANTERION and IOLMaster 700.

**Figure 3 diagnostics-12-00598-f003:**
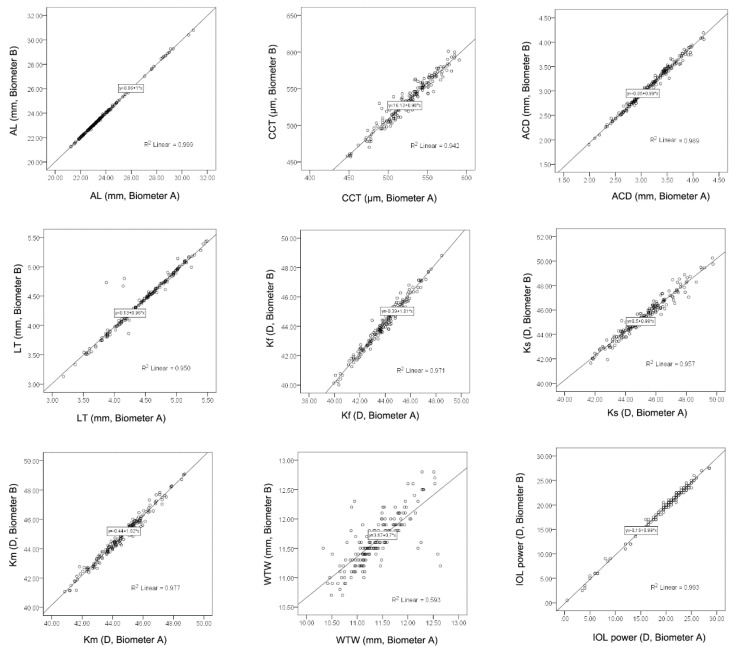
Scatterplots demonstrate the correlation between the axial length (AL), central corneal thickness (CCT), anterior chamber depth (ACD), lens thickness (LT), flat keratometry (Kf), steep keratometry (Ks), mean keratometry (Km), white-to-white (WTW) distance, and intraocular lens (IOL) power measured using biometers A and B. The regression equation between the two biometers is demonstrated in the rectangular box (x represents biometer A; y represents biometer B).

**Figure 4 diagnostics-12-00598-f004:**
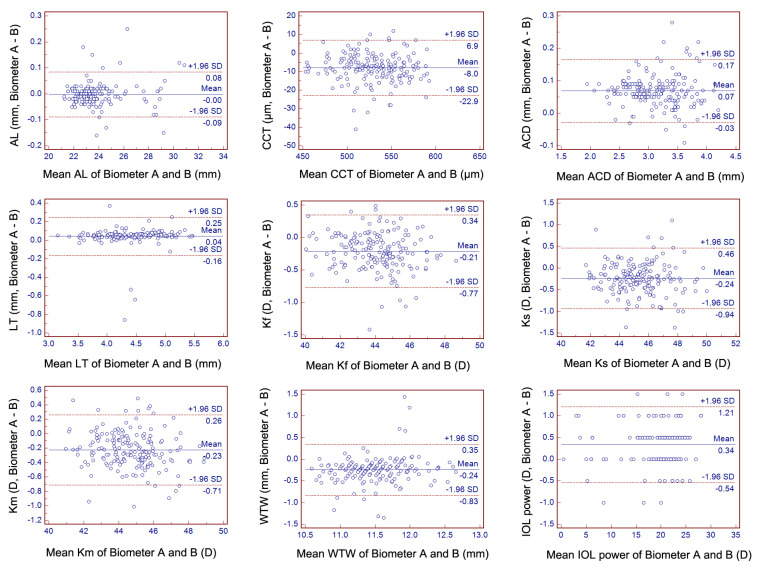
The agreement of axial length (AL), central corneal thickness (CCT), anterior chamber depth (ACD), lens thickness (LT), flat keratometry (Kf), steep keratometry (Ks), mean keratometry (Km), white-to-white (WTW) distance, and intraocular lens (IOL) power measured using biometers A and B (Bland–Altman plots). The mean difference is demonstrated with the continuous line, whereas the 95% limit of agreement is indicated with dashed lines.

**Table 1 diagnostics-12-00598-t001:** Anterior corneal astigmatism difference between the two devices.

	Biometer A (179 Eyes)	Biometer B (179 Eyes)	A–B	*p* *
Astigmatism magnitude (D)	1.25 ± 0.88	1.27 ± 0.85	−0.02 ± 0.41	0.439
Axis (degree)	75 ± 50	85 ± 56	−11 ± 54	**0.009 ^#^**
J0	−0.05 ± 0.51	−0.03 ± 0.57	−0.01 ± 0.82	0.841
J45	0.03 ± 0.57	0.03 ± 0.51	0.01 ± 0.75	0.902

Note: A = ANTERION; B = IOLMaster 700; D = diopter. * Calculated using paired sample *t*-test. ^#^ Values with a significance level of 5% are set in bold.

**Table 2 diagnostics-12-00598-t002:** Anterior segment parameters in both devices.

	Biometer A (179 Eyes)	Biometer B (179 Eyes)	A–B	*p* *
AL (mm)	23.71 ± 1.82	23.71 ± 1.82	−0.002 ± 0.04	0.469
CCT (μm)	526 ± 31	534 ± 32	−8 ± 8	**<0.001 ^#^**
ACD (mm)	3.19 ± 0.46	3.12 ± 0.46	0.07 ± 0.05	**<0.001 ^#^**
LT (mm)	4.46 ± 0.47	4.42 ± 0.46	0.04 ± 0.11	**<0.001 ^#^**
Kf (D)	43.96 ± 1.62	44.17 ± 1.67	−0.21 ± 0.28	**<0.001 ^#^**
Ks (D)	45.21 ± 1.68	45.44 ± 1.71	−0.24 ± 0.36	**<0.001 ^#^**
Km (D)	44.57 ± 1.59	44.80 ± 1.64	−0.23 ± 0.25	**<0.001 ^#^**
WTW (mm)	11.38 ± 0.46	11.62 ± 0.42	−0.24 ± 0.30	**<0.001 ^#^**
IOL power (D)	19.8 ± 5.2	19.5 ± 5.1	0.34 ± 0.45	**<0.001 ^#^**

Note: A = ANTERION; ACD = anterior chamber depth; AL = axial length; B = IOLMaster 700; CCT = central corneal thickness; D = diopter; IOL = intraocular lens; Kf = flat keratometry; Ks = steep keratometry; Km = mean keratometry; LT = lens thickness; WTW = white-to-white. * Calculated using paired sample *t*-test. ^#^ Values with a significance level of 5% are set in bold.

**Table 3 diagnostics-12-00598-t003:** The linear regression formulas and correlations for the anterior segment parameters in both devices.

	Linear Regression Formula	R^2^
AL (mm)	y = 0.06 + 1 × x	0.999
CCT (μm)	y = 16.13 + 0.98 × x	0.942
ACD (mm)	y = 0.05 + 0.99 × x	0.989
LT (mm)	y = 0.13 + 0.96 × x	0.950
Kf (D)	y = −0.39 + 1.01 × x	0.971
Ks (D)	y = 0.5 + 0.99 × x	0.957
Km (D)	y = −0.44 + 1.02 × x	0.977
WTW (mm)	y = 3.67 + 0.7 × x	0.593
IOL power (D)	y = 0.16 + 0.99 × x	0.993

Note: ACD = anterior chamber depth; AL = axial length; CCT = central corneal thickness; D = diopter; IOL = intraocular lens; Kf = flat keratometry; Ks = steep keratometry; Km = mean keratometry; LT = lens thickness; x = ANTERION; y = IOLMaster 700; WTW = white-to-white.

**Table 4 diagnostics-12-00598-t004:** Summary of the published ocular parameters differences between the two devices.

		Anterion—IOLMaster 700 Mean Difference								
	Sample Size	AL (mm)	CCT (μm)	ACD (mm)	LT (mm)	Kf (D)	Ks (D)	Km (D)	WTW (mm)	IOL Power (D)
Panthier C [7]	125	−0.01	−9	0.06	0.07	-	-	−0.11	−0.26	-
Shetty N [8]	127	−0.04	1.5	0.061	0.09	−0.15	−0.15	−0.15	−0.24	−0.19
Oh R [11]	47	−0.005 *	0.702	0.058	0.154 ^#^	−0.166	0.034	−0.059	-	-
Tañá-Sanz P [12]	102	-	−7.637	0.067	0.062	-	-	-	−0.149	-
Tañá-Rivero [13]	49	−0.0044	−6.8	0.0615	−0.0591	−0.0307	−0.0435	-	-	-
Fişuş AD [14]	389	−0.01	−5.66	0.07	0.06	−0.14	−0.11	−0.11	-	-
Pfaeffli OA [15]	78	−0.01	-	0.07	0.07	0.07	0.03	-	−0.22	-

Note: ACD = anterior chamber depth; AL = axial length; CCT = central corneal thickness; D = diopter; IOL = intraocular lens; Kf = flat keratometry; Ks = steep keratometry; Km = mean keratometry; LT = lens thickness; WTW = white-to-white. * Forty-one eyes for comparison; ^#^ 44 eyes for comparison; - = no reported data in paper; [ ] = the reference number sequence corresponding to a reference number in the paper.

## Data Availability

The data presented in this study are available in Appendix A.

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
