# Peer review of "Comparison Study of the Two Biometers Based on Swept-Source Optical Coherence Tomography Technology"

_diagnostics, 2022, doi:10.3390/diagnostics12030598_

Round 1
Reviewer 1 Report
In this manuscript, authors have compared various biometric measurements of eye using two commercially available two biometers: ANTERION and IOL Master 700. They compared a detailed parameters such as including axial length (AL), central corneal thickness (CCT), anterior chamber depth (ACD), lens thickness (LT), flat keratometry (Kf), steep keratometry (Ks), mean keratometry (Km), astigmatism, white-to-white (WTW) distance, acquired rate, and intraocular lens (IOL) power. ANTERION is operating at 1300 nm center wavelength, whereas IOL Master 700 has center wavelength of 1060 nm. All measurements were taken from 179 eyes. This is a interesting piece of study with clinical relevance and pointing out the credibility of the commercially available biometers to measure various physical and optical parameters of eye. Overall the manuscript is written well. However, it needs small amendments including brief explanations at various text positions prior to the final acceptance.
- Do authors think any difference in the biometric data if the eyes are dilated? Why authors preferred to measure without dilation. Need an explanation.
- Authors could mention a brief note on the technical reason for the small discrepancy (Interchangability) in all measures even within the confidence interval level. Measurement discrepancy between the systems are primarily caused by (i) difference in the operating wavelengths (ii) difference in the wavenumber calibration and dispersion compensation schemes adopted in the system [1-2]. [1] https://doi.org/10.1088/0031-9155/61/21/7652 [2] https://doi.org/10.1088/1612-2011/12/5/055601
- 1300 nm wavelength imaging is typically used for biometer/OCT imaging applications of anterior chamber primarily because of their strong absorption by aqueous humor and vitreous humor. On the other hand, 1060 nm has the least absorption relative to 1300 nm and nearly-zero water dispersion. This makes 1060 nm more preferred for deep retinal and choroidal imaging applications.
- Images in the Figure 4 look pixelated in my pdf file. Please verify it and use a high-quality graphics.
- It would be great if authors can show the images of both machines in the manuscript.
Author Response
In this manuscript, authors have compared various biometric measurements of eye using two commercially available two biometers: ANTERION and IOL Master 700. They compared a detailed parameters such as including axial length (AL), central corneal thickness (CCT), anterior chamber depth (ACD), lens thickness (LT), flat keratometry (Kf), steep keratometry (Ks), mean keratometry (Km), astigmatism, white-to-white (WTW) distance, acquired rate, and intraocular lens (IOL) power. ANTERION is operating at 1300 nm center wavelength, whereas IOL Master 700 has center wavelength of 1060 nm. All measurements were taken from 179 eyes. This is a interesting piece of study with clinical relevance and pointing out the credibility of the commercially available biometers to measure various physical and optical parameters of eye. Overall the manuscript is written well. However, it needs small amendments including brief explanations at various text positions prior to the final acceptance.
- Do authors think any difference in the biometric data if the eyes are dilated? Why authors preferred to measure without dilation. Need an explanation.
Answer: Thank you for your comments. We do believe that pupil dilation will affect the biometric data, which we have done the related research in our previous study using Lenstar LS 900 device (Wang X, Dong J, Tang M, Wang X, Wang H, Zhang S. Effect of pupil dilation on biometric measurements and intraocular lens power calculations in schoolchildren. PLoS One. 2018 Sep 13;13(9):e0203677.). As cataract surgeons, we really focus on the IOL power calculation. For routine IOL power calculation in clinic, the pupil dilation is not needed and we still believe that pupil dilation will affect the lens position and anterior chamber depth, which will influence the ELP and IOL power calculation.
- Authors could mention a brief note on the technical reason for the small discrepancy (Interchangability) in all measures even within the confidence interval level. Measurement discrepancy between the systems are primarily caused by (i) difference in the operating wavelengths (ii) difference in the wavenumber calibration and dispersion compensation schemes adopted in the system [1-2]. [1] https://doi.org/10.1088/0031-9155/61/21/7652 [2] https://doi.org/10.1088/1612-2011/12/5/055601
Answer: Thank you for your comments. We added the important information and corresponding references in the discussion part.
- 1300 nm wavelength imaging is typically used for biometer/OCT imaging applications of anterior chamber primarily because of their strong absorption by aqueous humor and vitreous humor. On the other hand, 1060 nm has the least absorption relative to 1300 nm and nearly-zero water dispersion. This makes 1060 nm more preferred for deep retinal and choroidal imaging applications.
Answer: Thank you for your comments. This may also be an explanation for the different acquired rates between the two devices. We also added this information in the discussion part.
- Images in the Figure 4 look pixelated in my pdf file. Please verify it and use a high-quality graphics.
Answer: Thank you for your comments. We make sure that the figures in the manuscript have a resolution of 300dpi.
- It would be great if authors can show the images of both machines in the manuscript.
Answer: Thank you for your comments. We added a figure (Figure 1) to show the two devices in the manuscript.

Reviewer 2 Report
This study compares two biometers based on SS-OCT and found ≥ 0.5 D difference in IOL power between them. It would be interesting to reveal which value was used in order to chose the IOLs to be implanted and what was the refractive outcome of surgery as well as its impact on the visual function and quality of life of the patients.
Author Response
This study compares two biometers based on SS-OCT and found ≥ 0.5 D difference in IOL power between them. It would be interesting to reveal which value was used in order to chose the IOLs to be implanted and what was the refractive outcome of surgery as well as its impact on the visual function and quality of life of the patients.
Answer: Thank you for your comments. We will do the post-operative refractive outcome follow-up in our next project to find the most influential parameters and to find which device showed a higher accuracy in terms of IOL power prediction in the clinic. We now also emphasize this limitation in the discussion part.

Round 2
Reviewer 2 Report
Without presenting the final refractive outcomes of the patients, I don't feel that the study is clinically relevant.
Author Response
Without presenting the final refractive outcomes of the patients, I don't feel that the study is clinically relevant.
Answer: We fully agree with the reviewer that refractive outcomes would add additional information to our paper. However, the focus of the article was rather to investigate the potential differences in the individual parameters itself, including axial length, central corneal thickness, anterior chamber depth, lens thickness, flat keratometry, steep keratometry, mean keratometry, astigmatism, white-to-white distance, acquired rate, and intraocular lens power, between the two swept-source optical coherence tomography biometers. Such data is – to the best of our knowledge – so far not available.
The final refractive outcome is not available and we therefore cannot provide it. But we still feel that very important and clinically relevant findings are presented. Most of the assessed parameters are not just used for IOL power calculation (e.g. axial length for strabismus surgery, white-to-white diameter for ICL size calculation, central corneal thickness in glaucoma patients etc.). Therefore, it is important for ophthalmologists to know in what regard the two new devices measure each parameter differently. From the published literature, it can certainly be expected that a ≥ 0.5 D difference in IOL power between the two biometers in 65.4% of the eyes will also have an impact on the refractive outcome. This could and should be evaluated in future studies.
Our goal was not to assess the impact on the refractive outcomes of patients. We mention the lack of such data as a limitation of our study. The sample size in our study was powered on axial length and would most likely not suffice to conclusively discuss refractive outcomes. However, we now discuss this topic based on a published article: Based on a previous study and conclusions, for every diopter of change in IOL power, only 0.7D of change in refraction at the spectacle level will be achieved (see reference below). Therefore, we used the formula: IOL prediction error = 0.7*IOL power error to find the potential difference between the two devices. This may also provide some useful information in clinic.
Reference:
- Feiz V, Mannis MJ, Garcia-Ferrer F, Kandavel G, Darlington JK, Kim E, Caspar J, Wang JL, Wang W. Intraocular lens power calculation after laser in situ keratomileusis for myopia and hyperopia: a standardized approach. Cornea. 2001; 20: 792–797.)

Round 3
Reviewer 2 Report
The authors clarified the raised questions.